# The Treatment of Periprosthetic Fracture Revision of the Humerus with “Bamboo Support” Structural Allograft Technique—Atrophic Non-Union of a Post-Operative Periprosthetic Fracture after Reverse Total Shoulder Arthroplasty: A Case Report

**DOI:** 10.3390/jcm13030825

**Published:** 2024-01-31

**Authors:** Hsien-Hao Chang, Joon-Ryul Lim, Tae-Hwan Yoon, Yong-Min Chun, Hyoung-Sik Kim

**Affiliations:** 1Department of Orthopaedic Surgery, Yongin Severance Hospital, Yonsei University College of Medicine, Seoul 03722, Republic of Korea; hsienchang@yuhs.ac; 2Department of Orthopaedic Surgery, Arthroscopy and Joint Research Institute, Severance Hospital, Yonsei University College of Medicine, Seoul 03722, Republic of Korea; 3Department of Orthopaedic Surgery, Gangnam Severance Hospital, Yonsei University College of Medicine, Seoul 03722, Republic of Korea

**Keywords:** periprosthetic fractures, periprosthetic humeral fractures, reverse total shoulder arthroplasty (RTSA), bamboo support technique

## Abstract

Periprosthetic fractures are a serious complication of joint replacement surgery. With the growing prevalence of reverse total shoulder arthroplasty (RTSA), the incidence of relatively uncommon periprosthetic humeral fractures has increased. Here, we present the unique case of a 74-year-old woman who developed atrophic non-union after plate osteosynthesis for a periprosthetic fracture associated with RTSA. Fixation failure was evident 3 months after the surgical intervention; the patient underwent a 3-month course of arm sling immobilization. However, bone resorption continued, and varus angulation of the fracture developed. In this case, surgical strategy involved the use of long proximal humerus internal locked system plate (DePuy Synthes, Paoli, PA, USA), augmented with autologous iliac bone graft and allogenic humerus structural bone graft with the “bamboo support technique”, fixed with Cable System (DePuy Synthes, Paoli, PA, USA). No reports have addressed the management of failed periprosthetic fractures using allogeneic humeral strut bone grafts. This report aims to fill the gap by presenting a novel surgical technique for the management of periprosthetic fractures associated with RTSA in case of treatment failure.

## 1. Introduction

The use of reverse total shoulder arthroplasty (RTSA) has expanded beyond rotator cuff tear arthropathy into a diverse range of conditions [1]. It is considered a viable treatment option for osteoarthritis, three- and four-part proximal humerus fractures, avascular necrosis of the humeral head, chronic locked dislocations, rheumatoid arthritis, failed anatomical shoulder arthroplasty, and oncologic conditions [2]. The emergence of periprosthetic fractures has presented a growing concern, primarily attributed to the increasing life expectancy and rising prevalence of RTSA surgery. Periprosthetic fractures after RTSA have a prevalence of 1% to 20% [3]. Whereas the incidence of periprosthetic fractures is increasing, they pose a substantial challenge in orthopedics, particularly for initial treatment failure.

The treatment method may vary depending on the fracture site (an initial periprosthetic fracture) or the non-union type (operative treatment of the initial periprosthetic fracture fails). Despite multiple classification systems for periprosthetic fractures associated with RTSA, there are limited surgical plans for periprosthetic fractures distal to stem implants [4,5,6]. A stable stem implant can be preserved, and the combination of a plate and cerclage wire construct for internal fixation is the preferred method [7,8]. Hypertrophic and atrophic non-union are two distinct types of non-union. Hypertrophic non-union is characterized by inadequate stability with adequate blood supply and biology. By contrast, atrophic non-union is caused by the lack of mechanical instability and blood supply to the fracture site. Regarding treatment, hypertrophic non-union responds to stabilization alone, whereas atrophic non-union typically requires both augmentation in stability and the biological stimulation with bone grafting to promote healing [9].

In this study, we present a unique successful post-operative surgically treated case utilizing an allogeneic humeral structural bone graft for the management of failed osteosynthesis in a periprosthetic fracture (Wright and Cofield Classification Type C) associated with RTSA [10]. This distinctive case highlights the importance of exploring innovative solutions in the face of challenging orthopedic scenarios. The utilization of an allogeneic humeral structural bone graft serves as a testament to the evolving nature of orthopedic interventions and the need for creative approaches when faced with failure of the initial treatment. Furthermore, the significance of this case extends to addressing the broader challenges associated with periprosthetic fractures after RTSA. The lack of comprehensive surgical plans for fractures distal to stem implants emphasizes the need for ongoing research and the development of standardized approaches. This case report serves as a valuable contribution to the evolving understanding of periprosthetic fractures, paving the way for future advancements in the treatment and management of these challenging orthopedic conditions. The incorporation of innovative techniques, so-called “bamboo support” techniques using allogeneic humeral structural bone grafts, provides a glimpse into the potential avenues for improving patient outcomes and addressing the complexities posed by periprosthetic fractures in the era of expanding RTSA applications.

## 2. Case Report

A 74-year-old woman arrived at our outpatient clinic (Yongin Severance Hospital, Yongin, Republic of Korea) complaining of weakened strength and pain in her right dominant arm. The patient also showed signs of severe limitation of motion, which was unable to be measured by the doctor due to pain. Her surgical history suggested that she had undergone RTSA 8 years previously. The patient had favorable clinical outcomes after arthroplasty until she experienced a periprosthetic fracture (Wright and Cofield Classification Type C) after slipping (Figure 1).

After undergoing osteosynthesis at a nearby orthopedic clinic 9 months before (Figure 2), the patient experienced persistent chronic pain, prompting her visit at our outpatient clinic during the follow-up. Apart from hypertension, she had no relevant medical history. Physical examination suggested no tenderness but showed severe limitations in the range of motion (ROM) due to pain.

Upon reviewing the simple radiographs captured 3 months after the operation, we observed breakage of the proximal wires; all screws had loosened without pulling out, callus formation was absent, and varus deformity was noticeable (Figure 3). The patient was treated conservatively with an abduction brace for 6 months post-operatively. On simple radiographs captured 6 months post-operatively, the fracture site displayed bone absorption and a noticeable progression of varus angulation (Figure 4). Radiographic findings confirmed atrophic non-union, necessitating a comprehensive approach to address both stability and maintain the structure of the anatomical biology through revision osteosynthesis. In order to discern the weakness reported by the patient, a cervical spine evaluation, electromyography (EMG), and nerve conduction velocity study (NCV) were conducted, and all results were confirmed to be within normal range.

The patient decided to proceed with the revisional operation. The patient was placed in the beach chair position under general anesthesia. The deltopectoral and anterolateral skin incisions that were previously made were connected for use. As the dissection was deepened, severe adhesion tissues were encountered and were subsequently adhesiolyzed. Diffuse metallosis of the soft tissues was observed, prompting debridement, and all previously implanted hardware was completely removed. We prepared the whole allogeneic humeral bone provided by the national-run organization called Korea Public Tissue Bank. The distal diaphysis of the whole allogenic humeral bone was cut in an axial direction in order to match the length of the patient’s humerus. At first, the length of the patient’s humerus was measured intraoperatively; then, the allogenic humeral bone was cut accordingly. In addition, the cancellous bone in the humeral head was harvested for grafting to fill the empty screw holes (Figure 5).

The hematoma and necrotic debris at the fracture site were removed via irrigation and massive curettage. The fracture gap was subsequently filled with an autologous iliac bone harvested from the ipsilateral anterior superior iliac spine. The 10-hole-long PHILOS plate was used in an inverted manner to optimize and maximize the number of distal locking screws. At the maximum capacity, we inserted four unicortical screws in the proximal region. In addition, an allogenic humeral structural bone graft was applied laterally to the humerus with a possible longest length, and three 1.7 mm cerclage wires with Orthopaedic Cable System (DePuy Synthes, Paoli, PA, USA) were used for the augmentation of stability (Figure 6).

The patient returned to our outpatient clinic every two weeks for a general check-up. Her arm was immobilized with a shoulder brace for six weeks. After the initial six weeks post-operation, she was permitted to engage in passive range of motion exercises (including forward flexion, abduction, and internal rotation) with tolerable pain. Any weight-bearing exercise or motions were not allowed until three months post-operation. The fracture healed six months after undergoing revision fracture fixation (Figure 7). She had been working as an office cleaner before the traumatic event and was able to return to work with high satisfaction of her shoulder status. She achieved an ROM of 170° forward flexion, 160° abduction, 40° external rotation, and a second lumbar vertebra level for internal rotation. Her visual analog scale (VAS) score for pain, American Shoulder and Elbow Surgeons (ASES) score, and Subjective Shoulder Value (SSV) score were 0, 85, and 90, respectively.

## 3. Discussion

Fracture management in general depends on a thorough understanding of the anatomy, biomechanics, and patient-specific factors. The shift from conventional fixation to advanced interventions, such as arthroplasty, has transformed orthopedic care, challenging clinicians to restore patient’s function and prevent complications. RTSA addresses complex shoulder pathology, and periprosthetic fracture revision complicates post-operative outcomes. This case report describes the complexities of managing a revised periprosthetic humeral fracture after a failed initial open reduction and internal fixation (ORIF) in a RTSA setting. The transition from conventional fracture management to revision surgery warrants a comprehensive exploration of the factors influencing the outcomes in this demanding clinical scenario.

As the average lifespan of the population increases, the number of RTSA cases has increased accordingly. This has led to many inevitable complications, with periprosthetic fractures being one of them, considered to be challenging [11]. Furthermore, when it comes to revisional cases arising from such periprosthetic fractures, there is no standardized approach or optimized method of treatment. Therefore, this case report—which aimed to present a novel technique called the “bamboo support” technique—was specifically designed for relatively uncommon cases that may become more prevalent in the future.

This report introduces a groundbreaking modification, termed the “bamboo support” technique, to address a periprosthetic fracture revision after RTSA. This innovative adaptation of the allograft sandwich method demonstrates the evolving nature of orthopedic interventions and offers a creative solution to the unprecedented challenges of a failed open reduction and internal fixation after RTSA. This modified technique has not been reported in similar cases, emphasizing its unique application. The term “bamboo support” encapsulates the resilience and structural support of the technique, adding strength and ingenuity to stabilize periprosthetic fractures. The subsequent sections delve into the intricacies of this approach and explore its feasibility, efficacy, and potential implications for future clinical practice.

Highlighting the pivotal role of the index surgery in ORIF of periprosthetic fractures in RTSA is crucial. In this case, the inadequacy of the applied plate length resulted in a shortened working length of the screw, thereby compromising its stability. The nature of a short oblique or transverse fracture, indicative of high energy trauma, implies concurrent damage to the adjacent soft tissues. We cannot overstate the critical importance of preserving the periosteum during index surgery to ensure optimal biological conditions. However, we adopted an anterolateral approach with a minimal incision, suggesting a compromise in soft tissue preservation. The resulting atrophic non-union underscores the significance of meticulous surgical techniques with adequate stability and the need to prioritize biological considerations during index surgery to prevent complications in subsequent revisions.

The focus of this operation was distilled into four primary points: fracture site management, selection of the fixation device, enlarged previous screw holes management, and consideration of additional augmentation method. First and foremost, the fracture site was managed using an autologous iliac bone graft with meticulous curettage of the necrotic debris. Secondly, a crucial aspect of the surgical approach revolved around the fixation device selection aiming for the maximum purchase of a lengthy plate and screws in the proximal stump adjacent to the fracture site. The challenge arose due to the presence of enlarged screw holes, complicating plate fixation. This complication was attributed to the limited availability of proximal bone stump space resulting from the presence of the humeral stem. To address both issues simultaneously, the PHILOS plate was utilized in a reverse manner, effectively resolving the two aforementioned problems.

The third issue was how to address the screw hole enlargement caused by screw loosening. Despite maximizing the use of autologous iliac bone graft, it was sufficient enough to fulfill the fracture site, leaving, however, an insufficient amount to cover the rest of the screw holes. Therefore, we resolved this by utilizing the cancellous bone from the head of an allogenic humeral bone to fill this area. This issue has provided valuable insights for surgeons who may encounter similar revisional periprosthetic fractures in the future. As was performed for the patient in this case at the previous hospital, delaying the timing of the surgery can exacerbate bone erosion, which can potentially leading to worse outcomes during subsequent surgical interventions. Therefore, once diagnosed with a non-union and surgical intervention is deemed necessary, it is considered the correct decision to perform surgery at the earliest possible time.

Lastly, considering the need for augmented stability beyond plate fixation, we had to make a strategic decision, particularly considering the restricted number of purchasable screws in the proximal stump owing to the previously implanted humeral stem. Due to the existing implant stem, there was a shortage of a viable bone stump in the proximal region for purchase of the screws. Additionally, in the diaphysis distal to the stem, the presence of enlarged screw holes limited the insertion of screws to a constrained number. Considering the insufficiency of stability with the conventional device (PHILOS plate), a decision had to be made to address this issue. To enhance stability, an innovative approach was taken by cutting the all-humeral bone axially and implementing the “bamboo support” technique through a strut graft. An additional cerclage wiring procedure was performed to further reinforce structural integrity.

Several alternative bone graft options were considered to enhance the stability of this operation. The use of an auto-nonvascularized fibular bone graft stands out as a viable choice, promoting osteogenesis and providing structural support. Alternatively, we explored the option of allogeneic endosteal bone grafts, leveraging the benefits of preserved bone quality without the need for autograft harvesting. Another viable alternative is the use of dual plating instead of allogeneic strut bones. This approach involves employing two plates to reinforce the stability at the fracture site, providing an alternative means of achieving necessary structural support. Each option was weighed carefully in the context of the patient’s specific condition and surgical requirements, ultimately guiding the decision-making process to ensure optimal stability and successful revision surgery.

This case study is subject to certain limitations inherent to its nature, with the absence of comparative analyses between or among groups. In addition, the follow-up period was relatively short, underscoring the need for an extended duration to assess the long-term impact and durability of the technique comprehensively. Furthermore, the broader context of an aging population, marked by an increased prevalence of arthroplasty procedures, contributes to the increased incidence of periprosthetic fractures. Acknowledging these limitations is crucial for a nuanced interpretation of our findings and underscores the ongoing need for extensive research to refine and validate the proposed surgical interventions.

## 4. Conclusions

As the global population of the elderly continues to expand, the prevalence of arthroplasty inevitably contributes to a surge in periprosthetic fractures. Addressing the challenges posed by non-union periprosthetic fractures requires the continual introduction of more effective and innovative surgical techniques and devices. The “bamboo support” technique emerges as a promising addition to the armamentarium of approaches for managing periprosthetic fracture revision. By offering a unique combination of structural support and adaptability, it exemplifies the ongoing commitment to advancing orthopedic interventions to meet the evolving demands of an aging demographic and to enhance the outcomes in the complex landscape of revision periprosthetic fracture surgery.

## Figures and Tables

**Figure 1 jcm-13-00825-f001:**
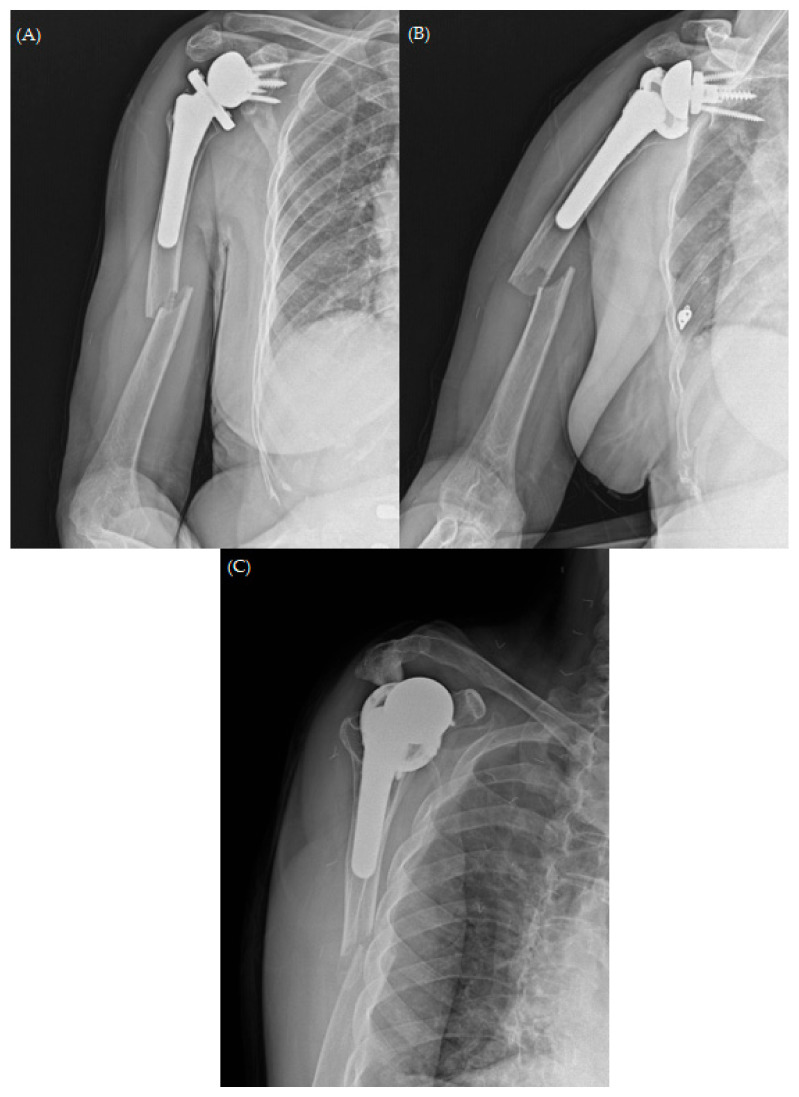
Plain radiographs of shoulder (**A**) anterior to posterior (AP), (**B**) outlet views, and (**C**) scapular Y-view demonstrating the short oblique type of periprosthetic fracture.

**Figure 2 jcm-13-00825-f002:**
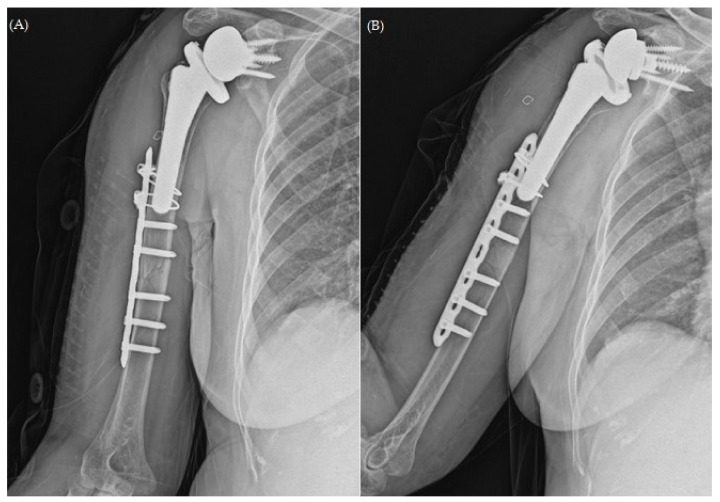
Plain radiographs of humerus (**A**) AP; (**B**) lateral views after open reduction and plate osteosynthesis for periprosthetic fracture.

**Figure 3 jcm-13-00825-f003:**
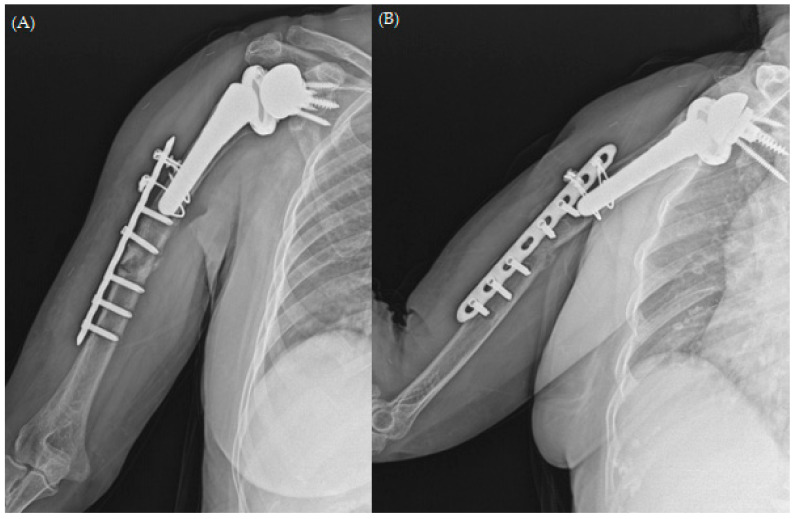
Plain radiographs of humerus (**A**) AP; (**B**) lateral views 3 months after open reduction and plate osteosynthesis for periprosthetic fracture.

**Figure 4 jcm-13-00825-f004:**
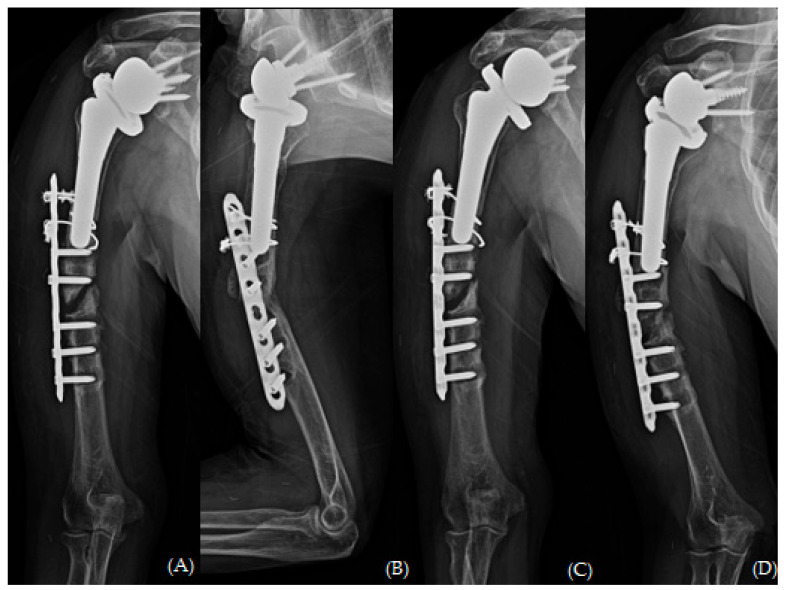
Plain radiographs of humerus (**A**) AP; (**B**) lateral and (**C**,**D**) oblique views 6 months after open reduction and plate osteosynthesis for periprosthetic fracture.

**Figure 5 jcm-13-00825-f005:**
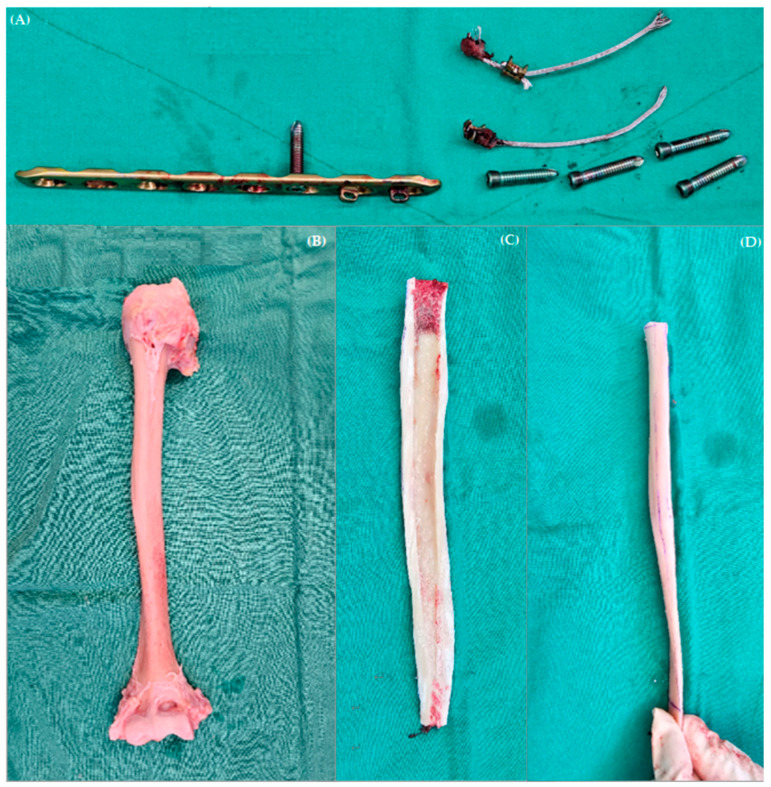
Intraoperative gross pictures of (**A**) removed previously inserted hardware, (**B**) allogenic humeral whole bone, and (**C**,**D**) prepared allogenic humerus whole bone for “bamboo graft”.

**Figure 6 jcm-13-00825-f006:**
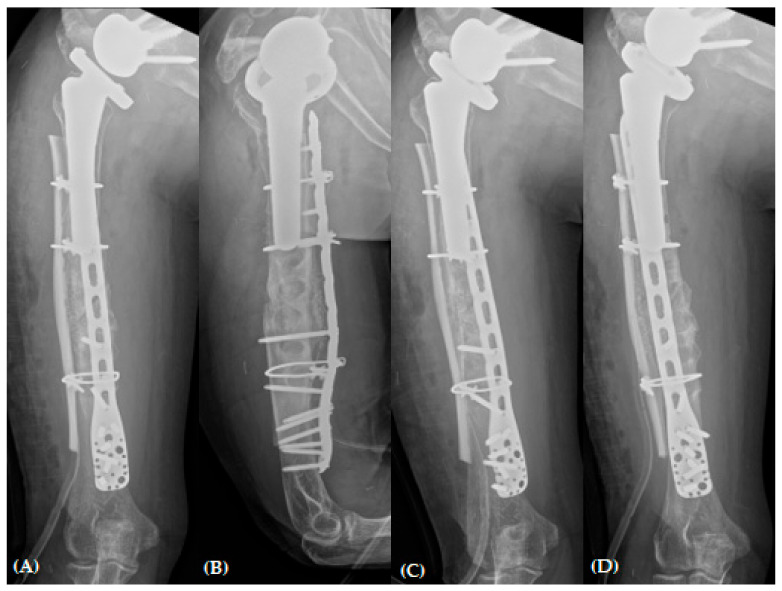
Plain radiographs of humerus (**A**) AP; (**B**) lateral and (**C**,**D**) oblique views immediately after revisional open reduction and internal fixation with “bamboo support” structural allograft technique.

**Figure 7 jcm-13-00825-f007:**
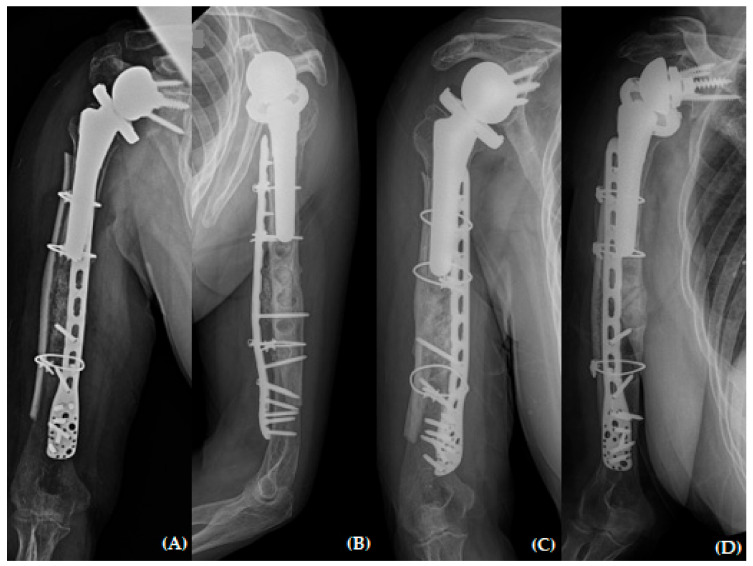
Plain radiographs of humerus (**A**) AP; (**B**) lateral and (**C**,**D**) oblique views 6 months after revisional open reduction and internal fixation with “bamboo support” structural allograft technique.

## Data Availability

Data are contained within the article.

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
