# Peer review of "The Treatment of Periprosthetic Fracture Revision of the Humerus with “Bamboo Support” Structural Allograft Technique—Atrophic Non-Union of a Post-Operative Periprosthetic Fracture after Reverse Total Shoulder Arthroplasty: A Case Report"

_jcm, 2024, doi:10.3390/jcm13030825_

Round 1

Reviewer 1 Report

Comments and Suggestions for Authors

The authors made the revision of the humerus fracture with “bamboo support” structural allograft technique, the research is interesting, however the manuscript need further information.

 The authors should be include all characteristics of the humerus whole bone donator (age, gender, blood type, etc. ) to understand why the patient body did not react with this insert.

 The authors should include how they determined or designed the spiral shape of the distal diaphysis of the humeral bone.

 The authors should explain what kind of proximal wires were used in the bamboo technique because in the first operation they were broken. The proximal wire were of the same type or they were more robust

Reviewer 2 Report

Comments and Suggestions for Authors

Dear authors,

Thank you for submitting your manuscript titled “Treatment of periprosthetic fracture revision of the humerus with “bamboo support” structural allograft technique—atrophic non-union of a postoperative periprosthetic fracture after reverse total shoulder arthroplasty: A Case Report and Surgical Technique” for review.

Despite the fact that it presents an innovative technique that can be useful in some clinical cases, the presentation does not have a logical thread, especially in the discussion part. I suggest revising the manuscript to improve the pitfalls presented. The final goal is to improve the overall clarity of the message to help the reader understand this interesting topic. Overall, my peer review is a major revision.

I have identified several areas where the manuscript could benefit from further enhancements. Below are my detailed suggestions:

-          Include the ethical approval number.

-          Use the JCM template

-          Introduction – please detail the classification systems for periprosthetic fractures associated with RTSA

-          Case report - The figures are eloquent but need to be further processed in order to group them.

-          Discussion – numerous own statements without bibliographic support.

§  Lines 164-167; 170-172; 218-220

o   There is redundant information in the discussion part and not presented in a logical manner.  It goes from one idea to another without connection. Some phrases are not related to contextual.

§  Lines 180-181; 182-183; 189-191

-          In the context of the limitations determined by the new technique and the lack of long-term studies, a comparison with other similar techniques would be useful – e.g “sarcophagus system”

-          I recommend to include several more recent (post-2021) references.

I hope these suggestions will be helpful in strengthening your manuscript and better conveying the important research you have undertaken. Looking forward to seeing the revised version of your work.

Best regards.

Reviewer 3 Report

Comments and Suggestions for Authors

In the manuscript, the authors report their experience in the case of a 74-year-old woman who developed atrophic non-union after plate osteosynthesis for a periprosthetic fracture associated with reverse total shoulder arthroplasty (RTSA). The surgical strategy involved using a long Proximal Humerus Internal Locked System plate, augmented with an autologous iliac bone graft and allogenic humerus structural bone graft with the “bamboo support technique.” The authors also underlined that this modified technique has not been reported in similar cases. Details of the novel surgical technique for managing periprosthetic fractures associated with RTSA were described in case of treatment failure. They adopted an anterolateral approach with a minimal incision, suggesting a compromise in soft tissue preservation. The surgical technique was summarized based on three primary focal points. In conclusion, the authors indicated that the “bamboo support” technique is a promising addition to the armamentarium of approaches for managing periprosthetic fracture revision. It underlines the role of this technique and can offer new directions for studies in the future.

Besides all the described details, additional information should be included, along with some corrections in different sections. The most important information I have underlined in the comments. Please find the attached document describing a few of the major issues noted in the manuscript.

Reference number of the manuscript: 2792952
REVIEWER COMMENTS
General comments
Point 1: I understand that there could be a prepared article in an accessible format and submitted, but I
suggest using a unique template that is indicated by the Journal of Clinical Medicine (JCM) in the
instruction for authors (the Microsoft Word template or LaTeX template) and prepare the manuscript;
this could be very helpful in the formatting of the sections and subsections.
Point 2: The titles are too long, and I suggest making some changes and shortening them because
everything is clear about the case report and surgical technique afterward.
Point 3: What is the period when this surgical technique was used?
Point 4: I suggest not using the term “our outpatient clinic” but stating details (name, place) of clinic (Line
64 and Line 74).
Point 5: What could be the disadvantages of this study? This point of view is essential and must be
explained in the general text.
Point 6: The sections “Materials and Methods” and “Results” are missing. Please prepare and add them to
the article. Some details from the section “Discussion” section should be included in the “Materials and
Methods” section.
Point 7: Is there any direct data on the range of motion (ROM) of a 74-year-old woman (Line 76) because
later in the text, she appears to achieve degrees of her in ROM (Line 152 and Line 153)? This could be
useful for comparison of the data.
Point 8: What types of biological factors are directly necessary through revision osteosynthesis (Line 94)?
Point 9: To what type of work did the patient return (Line 152)? Was there any rehabilitation before?
Point 10: Can any references provide their usage and information related to the American Shoulder, Elbow
Surgeons score, and Subjective Shoulder Value (Line 154 and Line 155)?
Point 11: Is the “Institutional Review Board Statement” really not applicable? The medical information of
1 patient was used, and there should be the Ethical permission to use this type of human data.
Point 12: I suggest the authors include more publications from the last 2 – 3 years in the text and the
references list because only one reference from the previous two years was mentioned there.
Point 13: The reference list related to the “Instructions for Authors” should be prepared and include
the abbreviated journal name, year, volume, page range, and DOI links.
Specific comments
Point 1: Please look at the keywords in Line 34: RTSA and reverse total shoulder arthroplasty are the same.
Why are these terms separated there?
Point 2: A full stop is missing at the end of the titles of Figure 1.
Point 3: A full stop is missing at the end of the titles of Figure 5. A comma is also missing between (A)
and (B) in Line 135.
Point 4: Please use “Conclusions” instead of “Conclusion” in Line 224.
Point 5: Look at the word “orthope-dic”, because there should be the word “orthopedic” in Line 231.

Reviewer 4 Report

Comments and Suggestions for Authors

Thank you for submitting your manuscript "Treatment of periprosthetic fracture revision of the humerus with "bamboo support" structural allograft technique - atrophic non-union of a postoperative periprosthetic fracture after reverse total shoulder arthroplasty: A Case Report and Surgical Technique" to Journal of Clinical Medicine.

Treatment of periprosthetic fracture revision of the humerus with "bamboo support" structural allograft technique—atrophic non-union of a postoperative periprosthetic fracture after reverse total shoulder arthroplasty: A Case Report and Surgical Technique

Thank you for submitting your manuscript "Treatment of periprosthetic fracture revision of the humerus with "bamboo support" structural allograft technique - atrophic non-union of a postoperative periprosthetic fracture after reverse total shoulder arthroplasty: A Case Report and Surgical Technique" to Journal of Clinical Medicine.

1.       Treatment of periprosthetic fracture revision of the humerus with "bamboo support" structural allograft technique - atrophic non-union of a postoperative periprosthetic fracture after reverse total shoulder arthroplasty: A Case Report and Surgical Technique - original case report article, is the theme of this manuscript, which is significant.

2.       This manuscript is a case report of a clinically exciting case.

3.       Complications after operatively treated shoulder fractures deserve attention, and the number of cases performed, and the treatment methods are few.

4.       X-ray imaging is well presented and described.

5.       It is good that the hypertrophic and atrophic types of stagnation and the different treatment methods are described separately.

6.       The work should focus on the specific case's advantages and disadvantages. It is unclear why this deserves attention; orthopaedics' operative technique is familiar. A good result alone does not justify a case report.

7.       In the text of lines 152-155, the functionality after the mentioned operation should be presented more concretely and comprehensively because the topic is also fascinating to specialists in other branches of medicine.

8.       The sentence in line 215 and 216 is not necessary.

9.       The structure of the manuscript is good.

10.    References are reasonable.

11.    This article is engaging and straightforward, but corrections are needed.

12.    There are many redundancies, missing determiners, and grammatical errors. English writing needs additional work.

13.    In conclusion, the treatment was successful because of union, stability, lack of early complications, and lack of medical problems.

* No ethical issues in this manuscript.

* No competing interest issues in this manuscript.

* No plagiarism issues in this manuscript were detected.

Comments on the Quality of English Language

1.      There are many redundancies, missing determiners, and grammatical errors. English writing needs additional work.

Author Response

Responses to the editor’s and reviewer 4’s comments

Title: Treatment of periprosthetic fracture revision of the humerus with “bamboo support” structural allograft technique—atrophic non-union of a postoperative periprosthetic fracture after reverse total shoulder arthroplasty: A Case Report and Surgical Technique

Type: Case study

Journal: Journal of Clinical Medicine

Summary:

We are grateful for the comprehensive review. We have attempted to revise our manuscript in accordance with the reviewers’ suggestion and comments. Our point-by-point responses to each of the reviewers’ comments are summarized below.

Point-by-point response to Comments and Suggestions for Authors

  1. Treatment of periprosthetic fracture revision of the humerus with "bamboo support" structural allograft technique - atrophic non-union of a postoperative periprosthetic fracture after reverse total shoulder arthroplasty: A Case Report and Surgical Technique - original case report article, is the theme of this manuscript, which is significant.

[Response]

Thank you for the comment.

  1. This manuscript is a case report of a clinically exciting case.

[Response]

Thank you for the comment.

  1. Complications after operatively treated shoulder fractures deserve attention, and the number of cases performed, and the treatment methods are few.

[Response]

Thank you for the comment.

  1. X-ray imaging is well presented and described.

[Response]

Thank you for the comment.

  1. It is good that the hypertrophic and atrophic types of stagnation and the different treatment methods are described separately.

[Response]

Thank you for the comment.

  1. The work should focus on the specific case's advantages and disadvantages. It is unclear why this deserves attention; orthopaedics' operative technique is familiar. A good result alone does not justify a case report.

[Response]

Thank you for the comment.

We the authors thought that this case is worth deserving attention due to its novelty in the methods of treating such fracture that has never been used before. We thought that the limitations were sufficiently stated in the last paragraph of the discussion section. If there are any points that need to be strengthened, please give us thorough suggestion so that we can improve our study for the better publishment.

  1. In the text of lines 152-155, the functionality after the mentioned operation should be presented more concretely and comprehensively because the topic is also fascinating to specialists in other branches of medicine.

[Response]

Thank you for the comment.

As other reviewers had commented in this part, we have revised as below. [lines 159-164]

  1. The sentence in line 215 and 216 is not necessary.

[Response]

Thank you for the comment.

As other reviewers had commented in this part, we have revised as below. [lines 222-223]

  1. The structure of the manuscript is good.

[Response]

Thank you for the comment.

  1. References are reasonable.

[Response]

Thank you for the comment.

  1. This article is engaging and straightforward, but corrections are needed.

[Response]

Thank you for the comment.

  1. There are many redundancies, missing determiners, and grammatical errors. English writing needs additional work.

[Response]

Thank you for the comment.

  1. In conclusion, the treatment was successful because of union, stability, lack of early complications, and lack of medical problems.

[Response]

Thank you for the comment.

  1. No ethical issues in this manuscript.

[Response]

Thank you for the comment.

This is revised and included.

  1. No competing interest issues in this manuscript.

[Response]

Thank you for the comment.

This is revised and included.

  1. No plagiarism issues in this manuscript were detected.

[Response]

Thank you for the comment.

Round 2

Reviewer 2 Report

Comments and Suggestions for Authors

Dear authors,

Thank you for re-submitting your manuscript titled “Treatment of periprosthetic fracture revision of the humerus with “bamboo support” structural allograft technique—atrophic non-union of a postoperative periprosthetic fracture after reverse total shoulder arthroplasty: A Case Report and Surgical Technique”.

There are some discrepancies between your answer and the submitted manuscript:

1.      You mentioned on lines 73-75 "The detailed classification for periprosthetic fractures associated with RTSA" - in fact the mentioned lines are included in the presentation of the case and the classification is not found in the text.

2.      The figures are not of a standard similar to the magazine's requirements. They are not grouped properly and the images are not marked either. For example, Figure 5 consists of 4 separate images without any indication.

3.      As long as there is no official document that certifies from an ethical point of view this case presentation and the procedure used, I consider that it was not approved.

Reviewer 3 Report

Comments and Suggestions for Authors

Dear Authors,

Points and suggestions of the general and specific comments in the article are improved and fixed from the authors' side. Besides, I have noted only a few minor issues (see comments).

Reference number of the manuscript: 2792952
REVIEWER COMMENTS
General comments
Point 1: Please look at the formatting of the numbers of the references in Line 48: without free places, i.e.
[7,8].
Point 2: In Line 177, the number of the reference should be placed before the sentence's punctuation: …
[10].
Point 3: Please look at the text formatting (spacing) between Lines 274 and 275.
Point 4: Please follow the instructions for formatting the references.
https://www.mdpi.com/journal/jcm/instructions
The reference list should include the full title, as the ACS style guide recommends. The abbreviations of
the journals in the indicated type, formatted in Italic style (Abbreviated Journal Name Year, Volume, page
range), and DOI links could be used in the reference list.
Point 5: The amount of references (especially from the last two years) could be increased.

Reviewer 4 Report

Comments and Suggestions for Authors

The revised version of manuscript is sufficient.

Author Response

Thank you for your comment. 

There was no request of revision for the Reviewer 4.